

# The concept of Watson's carative factors in nursing and their (dis)harmony with patient satisfaction

Majda Pajnkihar[1], Gregor Štiglic[1,2] and Dominika Vrbnjak[1]

[1] Faculty of Health Sciences, University of Maribor, Maribor, Slovenia
[2] Faculty of Electrical Engineering and Computer Science, University of Maribor, Maribor, Slovenia

## ABSTRACT

**Background**. Constant reviews of the caring behavior of nurses and patient satisfaction help to improve the quality of nursing. The aim of our research was to explore relationships between the level of nursing education, the perception of nurses and nursing assistants of Watson's carative factors, and patient satisfaction.

**Methods**. A questionnaire survey using a convenience sample of 1,098 members of nursing teams and a purposive sample of 1,123 patients in four health care institutions in Slovenia was conducted in August 2012. A demographic questionnaire and the Caring Nurse-Patient Interactions Scale (nurse version) were delivered to the nurses. A Hospital Consumer Assessment of Health Plans Survey was delivered to discharged patients. Data were analyzed using descriptive and inferential statistics.

**Results**. Carative factor sensibility was related to the level of nursing education. Patients were satisfied with the care received from nurses, nursing assistants and hospitals, although we found differences between the perceptions of nurses and nursing assistants of carative factors and patient satisfaction. By comparing only the perceptions of nurses and nursing assistants of carative factors in health care institutions, differences were found for seven out of ten carative factors.

**Discussion**. We did not find major significant differences between carative factors and level of nurse education, except in one carative factor. Differences in perceptions of carative factors between health care institutions are probably the result of different institutional factors. The results can be of great benefit to nurse administrators and educators, indicating the factors that must be taken into account for enhancing patient satisfaction. Emphasis on caring theories should be placed in nursing education and their application in nursing practice.

## INTRODUCTION

Caring is the core concept in nursing (*Brilowski & Wendler, 2005*; *Kyle, 1995*; *Pajnkihar, 2003*; *Palese et al., 2011*). Caring includes *caring for* and *caring about* clients (*Fisher & Tronto, 1990*; *Pajnkihar, 2003*). The first of these two main domains in holistic nursing is related to professional knowledge and expertise, and the second is related to the psychological and spiritual consideration of clients (*Pajnkihar, 2003*). One way to ensure that caring is central to the patients' experience is to endorse Watson's theory of human

Corresponding author
Majda Pajnkihar,
majda.pajnkihar@um.si

caring as the basis or a guide for nursing practice. To be able to perform a caring action, nurses need an artistic as well as a scientific knowledge and expertise (*Pajnkihar, 2008*). Nursing education plays an important role in the acquisition and advancement of caring attributes (*Labrague et al., 2015*) and this should be emphasized throughout their professional lives (*Pajnkihar, 2003*)

Caring behavior by nurses can contribute to the satisfaction and well-being of patients and is more than only the performance of the healthcare organizations (*Burt, 2007*; *Kaur, Sambasivan & Kumar, 2013*; *Sherwood, 1997*; *Wolf, Colahan & Costello, 1998*) or a specific type of professional and human-to-human contact. When caring is not present, non-caring consequences and dissatisfaction with care, where the person feels like an object, can occur. Caring has to be done in practice and research (*Watson, 2009*), as lack of caring is a major threat to health care quality. Because of the rapid advances in knowledge and technology, knowledge about care in practice must be constantly re-examined (*Pajnkihar, 2003*). A study by *Snowden et al. (2015)* could not confirm the association between previous caring experience by patients and higher emotional intelligence in nurses.

Regular reviews of nurses' caring behavior and actions as well as patient satisfaction may help nursing administrations to plan necessary improvement in practice. As there is a lack of research relating to the personal characteristics of nurses (such as education and caring behaviors) and a lack of empirical data on the relationships between carative factors as the core of caring and patient satisfaction in Slovenian health care institutions, we decided to explore these relationships.

## Carative factors as core of caring

Watson pointed out that caring is "the moral ideal of nursing whereby the end is protection, enhancement, and preservation of human dignity" (*Watson, 1999*, p. 29). Trustful and respectful interpersonal relationships are extremely important for preserving human dignity (*Pajnkihar, 2003*). Eriksson introduced the word 'carative' in caring science and defined it as love and charity and the motive for all caring (*Eriksson, 2006*). With her theory of caritative caring, she influenced Watson's work and development of carative factors (*Nelson & Watson, 2012*). Within Watson's theory, ten carative factors of love-heart-centered-caring/compassion represent the core of caring (*Watson, 2008*; *Watson, 2012*). Carative factors support and enhance the patients' caring experience (*Watson, 2008*). Watson's carative factors are seen as nurse-patient interactions and modalities that can be employed to support and enhance the experience of the actual caring occasion. These carative factors are described as consisting of: cultivating the practice of loving-kindness and equanimity toward self and others as foundational to caritas consciousness; being authentically present; enabling, sustaining and honoring the faith, hope and the deep belief system and the inner-subjective life world of the self and of the other; cultivating one's own spiritual practices and transpersonal self, going beyond the ego-self; developing and sustaining a helping-trusting, caring relationship; being present to, and supportive of, the expression of positive and negative feelings; creatively use the self and all ways of knowing as part of the caring process; engaging in the artistry of caritas nursing; engaging in genuine teaching-learning experiences that attend to the unity of being and subjective meaning;

attempting to stay within the other's frame of reference; creating a healing environment at all levels; administering sacred nursing acts of caring-healing by tending to basic human needs; opening and attending to spiritual or mysterious and existential unknowns of life and death (*Watson, 2008*). Watson later further developed carative factors into 10 caritas processes, described by *Jesse & Alligod (2014)*, where more consistent definitions of all caritas processes can be found. Carative factors and caritas processes facilitate healing, honour, wholeness and contribute to the evolution of humanity (*Watson, 2008*).

## Patient satisfaction as nurse sensitive patient outcome

Patient satisfaction is one of the established outcome indicators of the quality and the efficiency of healthcare systems (*Chow et al., 2009*; *Merkouris, Papathanassoglou & Lemonidou, 2004*) and also considered as a nursing-sensitive patient outcome, which is significantly impacted by nursing interventions (*Tervo-Heikkinen et al., 2008*). Satisfaction is related to patients' safety because it influences further health service utilization and the level of patients' adherence or compliance with prescribed treatments, regimens, and recommendations (*Palese et al., 2011*). Caring can be viewed as an antecedent or a consequence of safety in one's everyday life and health care (*Pajnkihar, 2003*). The interpersonal contact with nurses is central to patients' experience, hence the latter is a crucial determinant of the overall experience of care (*Merkouris, Papathanassoglou & Lemonidou, 2004*). A relationship was found between patient satisfaction and patient-perceived nurse caring (*Larrabee et al., 2004*; *Palese et al., 2011*). We can say that patient satisfaction is a response to nurse caring (*Sherwood, 1997*). However, there are many other related factors that can influence that perceptions.

## Level of nurse education in relation to their perception of caring and patient satisfaction

Nurse education should emphasize a holistic, individualized, and client-centred nursing care (*Pajnkihar, 2003*). Though it is also important, there is a lack of research relating caring perceptions to the personal characteristics of nurses, such as the level of education (*Patiraki et al., 2014*). The differences in patient satisfaction may be attributed to the different levels of educational background that nurses received in the participating countries (education spanning over a 3- and 4-year period, diploma and bachelor's levels) (*Palese et al., 2011*). Some researchers found no relationships between level of nurse education and patient satisfaction (*Patiraki et al., 2014*; *Weinberg, Cooney-Miner & Perloff, 2012*). Other results show that higher proportions of nurses with a bachelor's degree is important to achieve better patient outcomes and patient safety (*Aiken et al., 2013*; *Bruyneel et al., 2015*; *Weinberg, Cooney-Miner & Perloff, 2012*; *Weinberg et al., 2013*). The characteristics of nurses contribute to the care delivered in healthcare organizations (*Idvall et al., 2012*), however, further research is needed to explore the relationship of nurses' educational levels and patient outcomes (*Ridley, 2008*). In Slovenia, the nursing education system consists of a minimum 4-year secondary education (for nursing assistants), 3-years of first cycle Bologna higher education (for nurses with a diploma degree) and 2-years of second cycle Bologna higher education (for nurses with a master's degree). Our nursing educational

system has been reformed several times in the last decades (*Prosen, 2015*). We have had 2, 2.5 or 3-year higher vocational education (for nurses with associate degrees) and 3-year diploma level education (for nurses with diploma degrees). When referring to nurses in this research we mean nurses with an associate degree, diploma degree or master degree. Nursing assistants are those with 4-year secondary school vocational education. Currently we have 72% ($n = 12,387$) nursing assistants and 28% ($n = 4,871$) of nurses in Slovenia (*National Institute of Public Health of the Republic of Slovenia, 2014*). Nurses with diploma and master's degrees are both often involved in holistic patient care as team leaders at different levels of management. Nursing teams are made up of members of both groups. Nursing assistants work under the supervision of nurses performing different nursing procedures in the nursing process and are mainly responsible in assisting patients' daily life activities.

Nurses with diploma degree and master's degree are independent experts in nursing care and are responsible for nursing care and independently and autonomously perform nursing procedures and interventions in the nursing process. They are also involved in holistic patient care as team leaders at different levels of management.

Nurses with master's degree are also responsible for systematic monitoring of clinical work, managing teams to improve clinical practice, initiating changes and improvements, which are directly introduced into the process of nursing care.

The aim of this study was to explore the relationships between level of nurse education, nurses' and nursing assistants' perception of carative factors, and patient satisfaction in Slovene health care institutions. More specifically, this study aimed to:

(a) Examine the relationship between the level of nurse education and their perception of carative factors;

(b) Examine relationships between nurses' and nursing assistants' perception of carative factors and patient satisfaction in Slovene health care institutions;

(c) Examine the differences in nurses' and nursing assistants' perception of carative factors in Slovene health care institutions.

## METHODS

### Design

This study was conducted using a descriptive cross-sectional survey design.

### Sample and setting

Due to the large size of the potential target population for the study and accessibility of the selected hospitals, convenience sampling was used for recruiting nurses and nursing assistants providing bedside care. Nurses working in management were excluded from the study. We collected data from four different healthcare institutions in Slovenia, ranging from large university clinical centers to small general hospitals and different units. The reason for this choice was the large regional coverage achieved by including the two largest secondary healthcare level institutions and two hospitals from other regions. Despite the small number of institutions included in the study, their geographical distribution contribute to better generalizability of the results. Questionnaires were distributed to 1,098

nursing assistants and nurses, representing 29.84% of 3,680 nurses and nursing assistants working in these four health care institutions, and 11.68% of 9,404 nurses and nursing assistants in all Slovenian hospitals. The 613 questionnaires returned gave an overall response rate of 55.83%.

The same four health care institutions were used to collect data from patients who were discharged during the time of the study. Questionnaires were distributed to 1,123 patients and 475 questionnaires were returned, giving an overall response rate of 42.3%.

## Measures

Two questionnaires were used, one for nurses and nursing assistants and another for patients.

### Questionnaire for nurses and nursing assistants

The level of nursing education was measured at the individual level as previously used in the United States of America (*Weinberg, Cooney-Miner & Perloff, 2012*; *Weinberg et al., 2013*). Questions were adapted to the local environment.

The Instrument Caring Nurse-Patient Interactions Scale–nurse version (CNPI–nurse version), developed to measure caring and corresponding to the ten carative factors proposed by Watson's theory of human caring, was used (*Cossette et al., 2005*). The 70-item questionnaire uses a 1-to-5 Likert scale. Respondents circle the number best corresponding to their belief concerning the statement (where 1 represents *not at all*, 5 *extremely*), frequency (where 1 represents *almost never*, 5 *almost always*) and satisfaction (where 1 represents *very unsatisfied*, 5 *very satisfied*). Items are grouped in the following ten carative factors: Humanism (items 1–6); Hope (items 7–13); Sensibility (items 14–19); Helping relationship (items 20–26); Expression of emotions (items 27–32); Problem solving (items 33–38); Teaching (items 39–47); Environment (items 48–54); Needs (items 55–64); Spirituality (items 65–70) (*Cossette, 2006*; *Cossette et al., 2005*; *Cossette & Pepin, 2009*). The mean score was calculated for each of ten carative factors with an additional overall caring score that was calculated by averaging the individual scores. The reported Cronbach alpha coefficients between sub-scales varied from 0.73 to 0.91 in the original tool (*Cossette et al., 2005*).

Questionnaires were translated into the Slovene language by a professional translator. Translations were discussed in the group of six researchers and nursing experts, to assess content validity, acceptability and feasibility. Internal reliability using Cronbach's alpha was assessed in the present study for CNPI–nurse version and the corresponding values of reported alpha coefficients between sub-scales in the present study varied from 0.70 to 0.94.

### Questionnaire for patients

We used a modified Hospital Consumer Assessment of Health Plans Survey (H-CAHPS-survey) to measure patient satisfaction as used in previous research in the United States (*Weinberg, Cooney-Miner & Perloff, 2012*; *Weinberg et al., 2013*). We used questions related to satisfaction received from nurses during the hospital stay (three questions). For this part a 4-level scale was used with the following levels: 1-"never," 2-"sometimes," 3-"usually,"

4-"never." In addition, an item from the H-CAHPS survey was used to measure satisfaction with the hospital: "Using any number from 0 to 10, where 0 is *the worst hospital possible* and 10 is *the best hospital possible*, what number would you use to rate this hospital during your stay?" Translation and assessment of content validity, acceptability and feasibility were done in the same way as previously described for CNPI–nurse version.

## Data collection and ethical considerations

Data were collected in August 2012. All four health care institutions review boards gave written permission for the research. Ethics approval was obtained from the Ethics committee of University of Maribor, Faculty of Health Sciences (2159-2/2012/703-SN, dated 26.03.2016). Participants were informed about the study aims prior to administration of the questionnaires. The researchers handed out questionnaires to members of nursing teams in different units in the four different health care institutions, including nursing assistants and nurses. The completed questionnaires for nurses and nursing assistants were returned in a sealed box clearly identifiable in the ward. This box was regularly emptied by researchers. The questionnaires for patients were completed after leaving the ward. On the day of their discharge each patient received a questionnaire in an envelope. The completed questionnaires were sealed in return envelopes addressed to the principal investigator. Patients were assured they could refuse participation by not completing the questionnaire. Participation in the study was voluntary and anonymous. Responses were treated with full confidentiality. Details that might disclose the identity of the participants in the study were omitted. A response to the questionnaire was indicative of consent to participate.

## Data analysis

To assess the relation between 10 carative factors and education level, we ran logistic regression where participants were divided into two groups based on their level of education and used carative factors as predictors. The "no diploma" group included all nursing assistants ($n = 327$), while the "diploma group" included nurses ($n = 266$) to obtain a two-class output variable. After removal of samples with missing values for any predictor or output variable, we obtained a dataset with 269 participants in the "no diploma" group and 221 in the "diploma group." To check for multicollinearity of predictor variables, we employed a variance inflation factor (VIF) that was calculated for each variable to quantify the severity of multicollinearity. All VIF scores, ranging from 2.27 to 4.73 lie below the threshold of 10, to represent a high level multicollinearity (*Kutner, Nachtsheim & Neter, 2004*) Statistical significance was set at $p < 0.05$.

Analysis of variance (ANOVA) was used to examine the differences in mean values for patient satisfaction with nurses' and nursing assistants' care, carative factors and overall caring scores between the four health care institutions.

Descriptive analysis was employed for descriptions by participants of patient satisfaction and to visualize the differences between patient satisfaction and carative factors in the four different health care institutions. Mean values with corresponding 95% confidence intervals for comparing institutions were calculated for care received from nurses and nursing assistants, and overall patient satisfaction. To allow comparison on the same scale,

**Table 1  Demographic characteristics for members of nursing teams and patients.**

| Members of nursing teams | Responses |
| --- | --- |
| Education | |
| Nursing assistants, *n* (%) | 327 (53.3%) |
| Nurses, *n* (%) | 266 (43.3%) |
| Other, *n* (%) | 13 (2.3%) |
| Missing, *n* (%) | 7 (1.1%) |
| Workplace, proportion of nurses (vs. nursing assistants) | |
| Hospital 1, *n* (%) | 35 (5.7%) |
| Nurses, *n* (%) | 15 (46.9%) |
| Hospital 2, *n* (%) | 63 (10.3%) |
| Nurses, *n* (%) | 20 (33.9%) |
| UCC 1, *n* (%) | 314 (51.2%) |
| Nurses, *n* (%) | 142 (46.3%) |
| UCC 2, *n* (%) | 191 (31.2%) |
| Nurses, *n* (%) | 86 (46.5%) |
| Missing, *n* (%) | 10 (1.6%) |
| **Patients** | |
| Hospitalization | |
| Hospital 1, *n* (%) | 50 (10.5%) |
| Hospital 2, *n* (%) | 41 (8.6%) |
| UCC 1, *n* (%) | 217 (45.7%) |
| UCC 2, *n* (%) | 167 (35.2%) |
| Level of patients' education | |
| Primary education, *n* (%) | 75 (15.8%) |
| Secondary education, *n* (%) | 256 (53.9%) |
| Higher education, *n* (%) | 101 (21.2%) |
| Missing, *n* (%) | 43 (9.1%) |

mean values were linearly transformed for care received from nurses and nursing assistants (from 1–4 to 1–5 interval) and for overall patient satisfaction (from 1–10 to 1–5 interval). Data were analyzed using R, version 3.0.3 (http://cran.org).

## RESULTS

### Participants' descriptions

The majority of participants were nursing assistants (53.3%), followed by nurses (43.3%). Thirteen (2.3%) respondents had the other educational background and were excluded from further analysis. The majority worked in the university clinical center (UCC 1) (51.2%). The lowest proportion of nurses was found for Hospital 2 (33.9%).

The majority of patients were hospitalized and discharged from UCC 1 (45.8%). More than half of patients had secondary education (53.9%) (see Table 1).

**Table 2 Results for logistic regression describing the relation between 10 carative factors as predictors and education level as outcome.**

| Carative factor | Estimate | Std. error | z value | Pr(>|z|) |
|---|---|---|---|---|
| (Intercept) | −0.174 | 0.895 | −0.194 | 0.846 |
| Humanism | −0.061 | 0.286 | −0.214 | 0.830 |
| Hope | 0.058 | 0.198 | 0.295 | 0.768 |
| Sensibility | −0.537 | 0.239 | −2.250 | **0.025** |
| Helping relationship | −0.235 | 0.319 | −0.735 | 0.462 |
| Expression of emotions | 0.408 | 0.298 | 1.367 | 0.172 |
| Problem solving | 0.198 | 0.294 | 0.674 | 0.501 |
| Teaching | 0.213 | 0.319 | 0.667 | 0.504 |
| Environment | −0.346 | 0.347 | −0.998 | 0.318 |
| Needs | 0.517 | 0.325 | 1.589 | 0.112 |
| Spirituality | −0.259 | 0.217 | −1.196 | 0.232 |

**Table 3 Patient satisfaction with care received from nurses and nursing assistants.**

| Care received from nurses and nursing assistants | Mean | 95% CI | Std. error |
|---|---|---|---|
| 1. During this hospital stay, how often did nurses treat you with courtesy and respect? | 3.79 | 3.74–3.83 | 0.48 |
| 2. During this hospital stay, how often did nurses listen carefully to you? | 3.69 | 3.63–3.74 | 0.57 |
| 3. During this hospital stay, how often did nurses explain things in a way you could understand? | 3.67 | 3.61–3.73 | 0.62 |

## Relationship between level of nurses' education and their perception of carative factors

Multivariate logistic regression was used to identify the carative factors that are significantly related to the level of education. Table 2 shows that one of the 10 carative factors (i.e., sensibility) demonstrates a statistically significant association with the output variable. The negative $\beta$ coefficient for sensibility relates higher levels of this carative factor to a group of nurses with no diploma.

## Relationships between nurses' and nursing assistants' perception of carative factors and patient satisfaction in Slovene health care institutions

Mean values representing patient satisfaction with care from nurses and nursing assistants during the hospital stay can be seen in Table 3.

When comparing carative factors as perceived by nurses and nursing assistants and patient satisfaction with their care, differences were found for all carative factors, except in Hospital 1 for carative factors sensibility, problem solving, and spirituality. When comparing mean scores of patient satisfaction with care from nurses and nursing assistants (transformed mean scores ranged from 3.43 to 3.77) and overall caring score (transformed mean scores ranged from 4.27 to 4.52), differences were found in all four health care institutions (see Table 4).

**Table 4  Relationships between carative factors as perceived by nurses, nursing assistants and patients for satisfaction with their care.**

|  | Hospital 1 | | Hospital 2 | | UCC1 | | UCC2 | | *p* |
|---|---|---|---|---|---|---|---|---|---|
|  | **Mean** | **95% CI** | **Mean** | **95% CI** | **Mean** | **95% CI** | **Mean** | **95% CI** |  |
| Care from nurses, nursing assistants | 3.77 | 3.65–3.89 | 3.43 | 3.15–3.71 | 3.64 | 3.57–3.71 | 3.59 | 3.48–3.70 | 0.061 |
| Humanism | 4.50 | 4.34–4.67 | 4.36 | 4.18–4.54 | 4.43 | 4.36–4.50 | 4.54 | 4.47–4.61 | 0.102 |
| Hope | 4.56 | 4.38–4.74 | **4.66** | 4.37–4.96 | 4.51 | 4.44–4.59 | 4.66 | 4.59–4.72 | 0.090 |
| Sensibility | 3.88 | 3.65–4.11 | 4.15 | 3.98–4.31 | 4.03 | 3.95–4.11 | 4.28 | 4.19–4.38 | **<0.001** |
| Helping Relationship | 4.34 | 4.17–4.51 | 4.33 | 4.18–4.48 | 4.35 | 4.28–4.41 | 4.50 | 4.42–4.57 | **0.026** |
| Expression of Emotions | 4.19 | 3.99–4.39 | 4.34 | 4.19–4.50 | 4.25 | 4.17–4.32 | 4.49 | 4.36–4.61 | **0.003** |
| Problem Solving | 4.09 | 3.86–4.33 | 4.28 | 4.12–4.45 | 4.14 | 4.06–4.22 | 4.41 | 4.32–4.49 | **<0.001** |
| Teaching | 4.19 | 4.02–4.37 | 4.25 | 4.06–4.44 | 4.21 | 4.14–4.29 | 4.44 | 4.36–4.52 | **0.001** |
| Environment | 4.55 | 4.41–4.69 | 4.57 | 4.41–4.74 | 4.51 | 4.44–4.58 | 4.66 | 4.60–4.73 | **0.034** |
| Needs | **4.61** | 4.47–4.75 | 4.62 | 4.46–4.78 | **4.62** | 4.56–4.69 | **4.74** | 4.69–4.80 | 0.071 |
| Spirituality | 4.11 | 3.85–4.36 | 4.28 | 4.11–4.45 | 4.24 | 4.16–4.33 | 4.41 | 4.33–4.50 | **0.022** |
| Overall caring | 4.27 | 4.11–4.42 | 4.38 | 4.23–4.53 | 4.32 | 4.25–4.39 | 4.52 | 4.46–4.58 | **<0.001** |

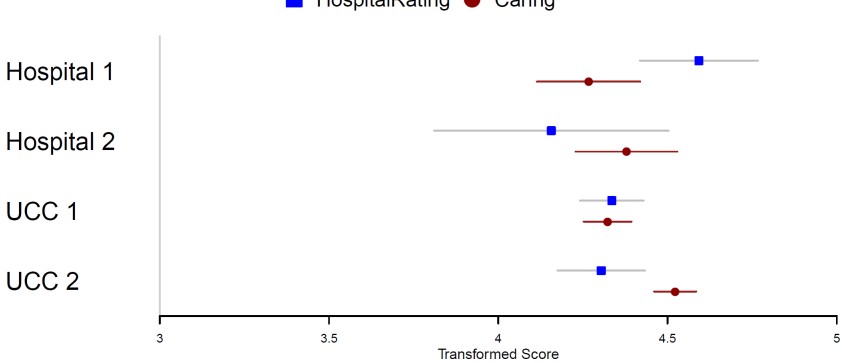

**Figure 1  Comparison of the patients' hospital satisfaction and overall caring score.**

When comparing overall patient satisfaction with the hospital and an overall caring score, differences were found only in UCC 2, where the mean for hospital rating was $M = 4.30$, 95% CI [4.18–4.33] and the mean for overall caring was $M = 4.52$, 95% CI [4.46–4.59] (see Fig. 1).

### Differences in nurses' and nursing assistants' perception of carative factors

When comparing nurses' and nursing assistants' perceptions of carative factors in four health care institutions, statistically significant differences were found for carative factors sensibility, helping relationship, the expression of emotions, problem solving, teaching, the environment and spirituality, and also for overall caring (see Table 4).

## DISCUSSION

This study aimed at exploring relationships between the level of nurses' education, nurses' and nursing assistants' perception of carative factors, and patient satisfaction in Slovene

health care institutions. We employed two validated questionnaires representing the two different constructs care (CNPI), and patient satisfaction (H-CAHPS).

The carative factor sensibility was related to the level of nurses' education. Higher levels of sensibility related to nursing assistants. Sensibility relates to being sensitive to self and others by nurturing individual beliefs, personal growth, and practices (*Watson, 2008*). Nurses should help people and be sensitive to themselves and others within a helping-trusting relationship that is directed towards protecting and enhancing their dignity (*Pajnkihar, 2003*). The results need to be addressed in practice, because in Slovenia nurses are responsible for theory-based nursing practice (*Kadivec et al., 2011*). Nurses should apply a nursing theory that contains elements of client-oriented nursing, holistic and systematic treatment and equal relationships between nurses and clients. Slovene nursing practice is mainly based on Virginia Henderson's conceptual model (*Pajnkihar, 2003*). Although Watson's theory has recently been implemented into educational programmes in Slovenia, students have accepted the theory positively and nurses in practice are increasingly aware of its importance (*Baznik, 2005*). Watson's theory as well as other nursing theories are now influencing clinical practice, especially as a framework for research in nursing. Henderson's model and Watson's theory have similiarities and differences. Among other authors that influenced the development of Watson's theory was also Virginia Henderson. Both Henderson and Watson believed that nursing should be holistic, stating that nursing is more than just meeting patients' physical needs, but also meeting psychosocial, social and spiritual needs. However, Henderson's model is more focused on helping, assisting patients' in their daily activities and meeting their needs, and Watson more on partnership caring relationships with individuals and focusing especially on spiritual aspects of individual lives. The results reflect the given practice that 72% of all members of nursing teams are nursing assistants who predominately carry out direct patient care, while only 28% (*National Institute of Public Health of the Republic of Slovenia, 2014*) of nurses are responsible not only for patient care but also for diagnostics and therapeutic treatment. This proportion shows the lack of nurses and the heavy workload of nurses probably influences their time spent in direct patient care. This trend should be reversed in our practice; we need 70% nurses and 30% nursing assistants (*Kadivec et al., 2011*). In our opinion there are not only educational factors that influence the perception of caring, but also that institutional factors influence the overall perception of caring.

Patients were satisfied with the care they received from nurses and nursing assistants (see Table 3), although we found some differences. Nurses and nursing assistants rated overall caring scores with higher mean scores than the patient participants did, and their satisfaction with care received by nurses and nursing assistants. Examining each carative factor separately, we found differences in nurses' and nursing assistants' perceptions of carative factors and patient satisfaction with their care for seven out of ten carative factors. There were no differences found for the three carative factors sensibility, problem solving and spirituality in one hospital (Hospital 1). Although we did not compare nurses', nursing assistants' and patients' perceptions toward nurses' caring behaviors, the findings are in line with the results of other authors who found that nurse participants rated caring behaviors with higher mean scores than patient participants did (*Omari,*

*AbualRub & Ayasreh, 2013*). There is considerable evidence that there is no congruence of perceptions between patients and nurses concerningwhich behaviors are considered caring and intended to be caring is not always perceived as such by the patient (*Papastavrou, Efstathiou & Charalambous, 2011*). Patient perceptions of important nurse caring behaviors differ from staff perceptions. Nurses perceive expressive/affective behaviors as being most important. In contrast, the patients perceive behaviors as most important when they demonstrate competent, instrumental know-how (*Von Essen & Sjoden, 2003*). This is also another possible explanation for the differences we found, as questions regarding care from nurses were related to courtesy, respect, listening and communication. Patients appear to value more instrumental and technical skills that nurses do (*Papastavrou, Efstathiou & Charalambous, 2011*). A study by *Bone (2008)* in maternity nursing also demonstrates the shift from the emotional connections between maternity nurses and women to the management and control of techno-medical interventions taking place. However, caring behaviors are important when performing diagnostic and therapeutic interventions. It is important to emphasize that patients were satisfied with the care they received from nurses and nursing assistants. Patients were also satisfied with the hospitals (see Fig. 1), but their satisfaction was not only dependent on nursing and care. When examining overall patient satisfaction with the hospital and overall caring, differences were found only in one university clinical center (UCC 2), where nurses and nursing assistants also estimated carative factors with higher mean scores than patient participants did. However, it should also be taken into consideration that comparison between the CNPI–nurse version and H-CAHPS survey was done on different Likert scales, which required linear transformation to allow comparison of mean values. A broad conceptual base must be adopted to understand phenomena in practice, because quality nursing and health care demand humanistic respect for the functional unity of the human being (*Pajnkihar, 2003*).

When examining differences in nurses' and nursing assistants' perception of carative factors in the four health care institutions, differences were found for the majority of carative factors and also for overall caring. In both clinical centers, as well as in one of the hospitals, the carative factor 'needs' received the highest average factor evaluated by nurses and nursing assistants. As education did not have such an impact on perceptions of carative factors, differences in health care institutions probably relate to different caring cultures and nurses' workload. This needs to be addressed in practice, especially by nursing administration. Some institutions adopt a caring-science paradigm, while the others should reproduce human caring through training that simulates caring (*Kay Hogan, 2013*). Also, stronger emphasis should be placed on caring theories during nursing education. Nurses need lifelong education and training to promote and maintain a vision, perspectives and values, and to assert that caring is essential in nursing (*Patiraki et al., 2014*). No differences were found for carative factors humanism, hope and needs. These elements seem to be common beliefs and values in Slovene nursing that are related to basic nursing philosophy and theoretical frameworks that underpin education and practice. Nursing needs lifelong learning of caring values and beliefs and constant interaction between theory and practice to support practice based on caring theories.

### Study limitations

There are some limitations to this study that should be taken into consideration when interpreting the results. A convenience sample of members of nursing teams and a purposive sample of patients were used to collect data, thus limiting the generalizability of our results. The construct validity of the patient satisfaction questionnaire was not tested due to small number of questions used. When measuring patient satisfaction, the questionnaire did not include information on specific nurses patients evaluated, thus limiting the possibilities of paired sample statistical tests. Due to the high number of questions in the CNPI scale, the motivation of the participants in the study might have been reduced, and the survey itself was time consuming for both researchers and participants.

## CONCLUSION

The essence of nursing is caring for others and for oneself. Nurses should be caring and be able to establish personal contact, have a communicational approach and good interpersonal relationships with patients. Interventions should reflect professionalism and care for individual problems, not only be seen as routine work (*Pajnkihar, 2003*). Patients appreciate good care, hence their perception about caring and satisfaction should be taken into consideration. As described by *Douglas (2010)*, caring is important for the well-being of staff working with patients. He concludes that delivering care without caring is wrong.

The environment and culture in health care institutions seems to have an important impact on the nurses' perception of carative factors. We echo *Watson* (*1999*, p. 75) that promotion of a "supportive, protective, and/or corrective mental, physical, societal, and spiritual environment" empowers nurses in caring partnerships with individuals (*Pajnkihar, 2003*). However, nurses also need time and resources to establish such caring environments. A stronger emphasis on caring theories is necessary during both nursing education and lifelong learning so that a strong core of knowledge relating to caring theories and theory based practice can exist.

Further research is needed to determine the influence of carative factors on patient satisfaction in longitudinal studies. In addition, different factors of patient satisfaction should be explored in detail. For this to happen, a mixed methods approach should be used. It would be useful to explore how the time constraints on nurses impact their interactions with their patients.

## ACKNOWLEDGEMENTS

The study was undertaken as a part of a bilateral research project 'Safety and caring for patients in a clinical environment' in correlation with the education of nurses between the University of Maribor Faculty of Health Sciences and IM Sechenov First Moscow State Medical University of the Ministry of Health and Social Development, Moscow. We thank the researchers Klavdija Čuček Trifkovič, Barbara Donik, Sabina Fijan, Barbara Kegl, Mateja Lorber, Maja Strauss, Jadranka Stričević, Sonja Šostar Turk, Natalya N. Kamynina, Alexey Y. Brazhnikov, Irina V. Ostrovskaya, Valentina E. Efremova and Igor S. Lunkov.

## Funding

This study was funded by the Slovenian Research Agency under the grant number BI-RU/12-13-036. There was no additional external funding received for this study. The funders had no role in study design, data collection and analysis, decision to publish, or preparation of the manuscript.

## Grant Disclosures

The following grant information was disclosed by the authors:
Slovenian Research Agency under the grant: BI-RU/12-13-036.

## Competing Interests

The authors declare there are no competing interests.

## Author Contributions

- Majda Pajnkihar, Gregor Štiglic and Dominika Vrbnjak conceived and designed the experiments, performed the experiments, analyzed the data, contributed reagents/materials/analysis tools, wrote the paper, prepared figures and/or tables, reviewed drafts of the paper.

## Human Ethics

The following information was supplied relating to ethical approvals (i.e., approving body and any reference numbers):

All four health care institutions review boards provided a written permission for research. Institional Review Boards from 4 institutions (595/1-1-2012, dated 1.6.2012; 0104-14/2012:2, dated 22.5.2012; no number, dated 27.8.2012; E 33/12-MB, dated 30.5.2012).

Ethics approval was obtained from the Ethics committee of University of Maribor Faculty of Health Sciences (2159-2/2012/703-SN, dated 26.03.2016).

All files are uploaded as confidental Supplemental File.

## Data Availability

The raw data has been supplied as a Supplementary File.

## Supplemental Information

Supplemental information for this article can be found online at http://dx.doi.org/10.7717/peerj.2940#supplemental-information.

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
