# Peer review of "The concept of Watson’s carative factors in nursing and their (dis)harmony with patient satisfaction"

_PeerJ, doi:10.7717/peerj.2940_

## Round 0.1 · original submission · Major Revisions

Thank you for this interesting article on caring and patient satisfaction. It would be of benefit to describe why you chose Watson's theory when you identify that Virginia Henderson's theory is the basis of nursing care in Slovenia. Editing of the manuscript for proper verbiage and grammar would be helpful. Addressing the comments from the reviewers is important for your revision of this manuscript. Based on the comments I feel that a major revision is warranted.

·

Basic reporting

There are some tense and sentence structure issues throughout the manuscript. The opening sentence of the abstract is not validated/supported by the literature.

Experimental design

No commnets

Validity of the findings

Table 1 unclear. Workplace proportion of nurse. For the hospital data, unclear as to what the % represents. Clarify the level of education ---care givers or patients.
Table 3-explanation could be more descriptive with regard to scale used to measure satisfaction.
Data in table 4- could is be displayed better visually/graphically. Too busy to decipher cleanly and quickly. The point it is trying to make gets lost.

Comments for the author

The explanation of the carative factors of Watson's theory could have been more in-depth. That would have given the readership a better understanding of the behaviors/concepts you were looking at.

·

Basic reporting

The paper is interesting and reports on the methods and results of a questionnaire survey on nurses’ and patients perception of caring and the role of education. The theoretical underpinning to the paper is Watson’s carative factors and patient satisfaction. The survey methods and results are well supported by tables presenting research instruments and results. The paper needs proof reading and some re-structuring especially in relation to paragraph construction to assist the reader in following the detailed information being presented. The text, tables and figures also need to be closely aligned. Some additional references have been suggested (see line 58 and 59 for example) on the role of education and its relationship to nurses’ personal characteristics and caring behaviours (e.g. Snowden A , Stenhouse R et al [2015] The relationship between emotional intelligence, previous caring experience and mindful ness in student nurses and midwives: a cross sectional analysis, Nurse Education Today, 35, 152-158)

Experimental design

The authors need to explain the rationale for their convenience sample to recruit nurses and their purposive sample to recruit patients and their choice of institutions. Issues of validity and generalizability of the findings need to be considered in the design.

Validity of the findings

Validity of findings
It would be helpful if the authors could comment on the construct validity of the questionnaires in relation to nurses’ and patients’ perceptions of carative factors – in other words are the Caring Nurse-Patient Interactions Scale and the Hospital Consumer Assessment of Health Plans Survey measuring the same construct?
Could this be one reason for the lack of congruence between nurse participants’ and patients’ perceptions (see Lines 294-297)
Line 97 - Patient satisfaction is assumed to be a response to nurses’ caring but other factors related to the patient could be influencing their perceptions.
Lines 119-221 - There is a huge difference between the number of nursing assistants and number of nurses taking part in the survey which reflects the nursing workforce in Slovenia. The carative factor ‘sensibility (Line 269) was shown to differ between the two groups suggesting that the level of education was the reason for this. The role of institutional factors and how they impact on nurses’ and nursing assistants' individual characteristics is an important discussion point. Could the authors speculate on these possible factors using the institutions where differences were observed as exemplars? (See lines 290-291 and 308)
The authors may be interested in research undertaken to show the influence of different levels of nurse education on nursing performance described in Wile A et al (1997) Measuring clinical nurse performance: development of the King's Nurse Performance Scale. International Journal of Nursing Studies 34 (3):222-230
Line 277 - The authors state that nursing practice in Slovenia is based on Virginia Henderson’s theory. This needs referencing and a brief explanation of the theory and how it both differs and is similar to Watson’s theory. This is important for the validity of the findings.
Lines 302-304 refer to the value patients place on the different types of caring (technical/instrumental over expressive/affective care). In the light of this finding the authors may find the following paper of interest (Bone 2008 Epidurals not Emotions: the care deficit in US maternity care. In: Hunter B and Deery R (eds) Emotions in Midwifery and Reproduction. Basingstoke, Palgrave Macmillan). The paper also highlights the negative influence of financial constraints on shaping modern health care which may be relevant to interpreting the findings
On Line 339 the authors state ‘Delivering care without caring is wrong’ (citing Douglas 2010). This is a fairly blunt statement and I would suggest the authors rewrite this sentence in a more nuanced way.

Comments for the author

This is an interesting paper and shows the importance of quantitative survey research for describing conceptually complex issues and flagging up research questions for further research using mixed methods. These points could be emphasised further throughout and especially in the conclusion.
The paper needs proof reading throughout.

·

Basic reporting

There are a number of English language grammar and sentence structure issues throughout the manuscript. Especially in the abstract, and also with the incorrect use of "manly" instead of "Mainly" (lines 277 and 279).

At times it is difficult to follow what is being stated due to difficulty with wording of the sentence or content. For example, in the abstract under "Results" the author states Carative factor sensibility was related to the level of nursing education. But then under the abstract "Discussion" the author states that she did not find major significant differences between carative factors and level of nursing education.

Experimental design

Difficult to understand how the level of nursing education and patient satisfaction of care are linked/correlated since the patient surveys were anonymous. How did the research team know what staff cared for the patient and what that nurses level of education was. This would have been an important factor to track given research aim b).

The Caring Nurse-Patient Interaction Scale - nurse version is a very long survey consisting of 70 questions potentially contributing to survey fatigue.

Validity of the findings

Somewhat small convenience sample size.
The nurse educational system in Slovenia is difficult to understand for a nurse practicing in the US. Do Master level nurses provide bedside care?

The point about differences in nurses perception of their caring and a patient's perception of a nurse's caring is clear. It is more difficult to track the educational preparation piece throughout the article, although it is addressed well in the Introduction, the Data Analysis does not really confirm this as it speaks to surveys on nurses' perception of caring rather than on what their nursing curriculum actually taught.

It is probably a valid point about the time constraints on nurses impacting on their interactions with their patients, but again, this was not addressed in the surveys that were administered. This would be a terrific area for further research!

Tables were clear and well organized.
One reference Burt, 2007 does not appear on the reference list.

Comments for the author

A very interesting and salient topic! Glad to see some work being done in this area.
English proofreading would be of a great benefit to this article.

---

## Round 0.2 · accepted · Accept

Thank you for your updated version of the manuscript. You addressed well the comments of the reviewers.

·

Basic reporting

Very nice job with the revision. Well worded and much easier to understand.
Great job in an area where we certainly need more research completed.

Experimental design

Good. Addressed limitations well in that section.

Validity of the findings

No concerns. Small, convenient sample. Results so much easier to comprehend and process now.

Comments for the author

Great job! Thank you!